# Evaluation of the Accuracy of the Aerosol Optical and Microphysical Retrievals by the GRASP Algorithm from Combined Measurements of a Polarized Sun-Sky-Lunar Photometer and a Three-Wavelength Elastic Lidar

Daniel Camilo Fortunato dos Santos Oliveira [1], Michaël Sicard [1,2], Alejandro Rodríguez-Gómez [1,*], Adolfo Comerón [1], Constantino Muñoz-Porcar [1], Cristina Gil-Díaz [1], Simone Lolli [1,3], Oleg Dubovik [4], Anton Lopatin [5], Milagros Estefanía Herrera [5] and Marcos Herreras-Giralda [5]

[1] CommSensLab, Department of Signal Theory and Communications, Universitat Politècnica de Catalunya, 08034 Barcelona, Spain; daniel.camilo.fortunato@upc.edu (D.C.F.d.S.O.); michael.sicard@univ-reunion.fr (M.S.); comeron@tsc.upc.edu (A.C.); constantino.munoz@upc.edu (C.M.-P.); cristina.gil.diaz@upc.edu (C.G.-D.); simone.lolli@upc.edu (S.L.)

[2] Laboratoire de l'Atmosphère et des Cyclones, Université de la Réunion, 97744 Saint-Denis, France

[3] CNR-Institute of Methodologies for Environmental Analysis (IMAA), Contrada S. Loja, 85050 Tito Scalo, Italy

[4] CNRS, UMR 8518—LOA—Laboratoire d'Optique Atmosphérique (LOA), Université de Lille, 59650 Lille, France; oleg.Dubovik@univ-lille.fr

[5] GRASP-SAS, Generalized Retrieval of Atmosphere and Surface Properties, 59260 Lezennes, France; anton.lopatin@grasp-sas.com (A.L.); milagros.herrera@grasp-sas.com (M.E.H.); marcos.herreras@grasp-sas.com (M.H.-G.)

* Correspondence: alejandro.rodriguez.gomez@upc.edu

**Abstract:** The versatile Generalized Retrieval of Aerosol and Surface Properties (GRASP) algorithm exploits the advantages of synergic ground-based aerosol observations such as radiometric (sensitive to columnar aerosol optical and microphysical properties) and lidar (sensitive to vertical distribution of the optical properties) observations. The synergy is possible when the complementary data is mutually constrained by GRASP parametrization that includes, for the first time ever, the degree of linear polarization (DoLP) parameter measured by a polarized sun-sky-lunar AERONET photometer (380, 440, 500, 675, 870, 1020, and 1640 nm) in synergy with the vertical profiles from an elastic lidar (355, 532, and 1064 nm). First, a series of numerical tests is performed using simulated data generated using a climatology of data and ground-based measurements. The inversions are performed with and without random noise for five different combinations of input data, starting from the AERONET-like dataset and increasing to the complex one by adding more information for three aerosol scenarios: I—high aerosol optical depth (AOD) with dominant coarse mode; II—low AOD with dominant coarse mode; III—high AOD with dominant fine mode. The inclusion of DoLP improves (i) the retrieval accuracy of the fine-mode properties when it is not dominant; (ii) the retrieval accuracy of the coarse-mode properties at longer wavelengths and that of the fine-mode properties at shorter wavelengths; (iii) the retrieval accuracy of the coarse-mode real part of the refractive index (up to 36% reduction), but has no effect on the retrieval of the imaginary part; (iv) reduces up to 83% the bias of the sphere fraction (SF) retrieval in coarse-mode dominated regimes; and (v) the root mean square error (RMSE) of the retrieval for most of the parameters in all scenarios. In addition, the addition of more photometer channels in synergy with a three-wavelength elastic lidar reduces the RMSE for the real part (67% in the coarse mode) and the imaginary part (35% in the fine mode) of the refractive index, the single scattering albedo (38% in the fine mode), the lidar ratio (20% in the coarse mode), and the SF (43%).

**Keywords:** GRASP algorithm; DoLP; synergy; photometer; lidar

## 1. Introduction

Aerosols play an essential role in climate because the particles scatter and absorb solar and terrestrial radiation and because they are the nuclei upon which cloud droplets and ice particles form and aggregate into cloud structures known to dominate the Earth's albedo [1]. The scattered light caused by some of these particles in the atmosphere is affected by components that affect not only intensity but also the state of scattered light [2]. Therefore, observations with polarimetric capabilities can provide improvements in the retrieval accuracy of aerosol properties [3–7]. The polarization information, sensitive to particle shape and index of refraction, in the inversion processes seems to be a promising complement for obtaining the properties of the aerosols and understanding their local and global contribution to climate, environmental, and human health.

In recent decades, efforts to improve the retrieval of the microphysical and optical properties of aerosols using synergies between remote sensing instruments have become more popular in the scientific community. Since the 2000s, several algorithms have been developed to enhance the inversions of aerosol properties by combining synergistically photometer and lidar data. The first combination between lidar and radiometer from AErosol RObotic NETwork (AERONET) was proposed by [8] to retrieve the optical properties of the altitude-inhomogeneous aerosol layer. Chaikovsky's approach paved the way for the development of other lidar-radiometer synergetic algorithms [9,10] and was consolidated and named LIdar-Radiometer Inversion Code (LIRIC) for processing data from European AErosol Research LIdar NETwork (EARLINET) measurements and disseminating the code in the network [10]. Likewise, Cuesta et al. (2008) developed the LidAlm (Lidar and Almucantar) synergetic technique to characterize the atmospheric column by its different aerosol layers, combining standard elastic backscatter lidar and inversion from an AERONET sun photometer [11]. In addition, Ganguly et al. (2009) demonstrated a technique to derive the concentration of aerosol components from AERONET and MPLNET (Micro-pulse Lidar Network) measurements [12].

Another algorithm developed by [13] is Polarization Lidar Photometer Networking (POLIPHON), taking its roots from the method of [14], to separate coarse-mode and fine-mode particle properties using information from AERONET and lidar data. Finally, the Generalized Aerosol Retrieval from Radiometer and Lidar Combined data (GARRLiC) approach, which is a branch of the Generalized Retrieval of Atmosphere and Surface Properties (GRASP) algorithm, inverts coincident multi-spectral lidar and radiometer observations and derives a united set of aerosol parameters [15]. The GARRLiC/GRASP algorithm benefits from the lidar sensitivity to the vertical properties of aerosols and the sensitivity of radiometric observations to the amount and type of aerosols in order to derive an extended set of parameters modeling aerosols as a bicomponent mixture. In addition, GRASP code is a versatile algorithm for retrieving a variety of atmospheric properties from diverse remote sensing observations, including passive and active measurements from space, ground, and aircraft [16]. It uses overall the same retrieval concept as the one developed for the AERONET algorithm [5,17–20] that was initially extended by [21] for retrieving aerosol properties from measurements by satellite polarimeter POLDER (POLarization and Directionality of the Earth's Reflectances version 3/Polarization and Anisotropy of Reflectances for Atmospheric Sciences coupled with Observations from a Lidar) and later to many other applications [22]. In this regard, the evolution of the GARRLiC/GRASP concept and new options of analyzing lidar data in the framework of GRASP have been described by [23].

Several simulation tests and applications of GRASP algorithm to real data have been conducted to check its capabilities for atmospheric and surface properties. Lopatin et al. (2013) explored the scope and accuracy of the aerosol retrieval from coincident lidar and sun photometer observations using GARRLiC/GRASP [15]. Torres et al. (2014) investigated the dependency of aerosol retrieval on the geometry used in the sky radiance measurements [24]. Fedarenka et al. (2016) investigated the improvement of aerosol retrieval by adding polarization measurements to an adapted AERONET code [7]. Derimian et al. (2016)

evaluated the radiative effect of several key aerosol types and the diurnal dependence on solar zenith angles of the aerosol radiative effect and for examining the effects of assumptions on the data [25]. Torres et al. (2017) developed GRASP/AOD retrieval, where a bimodal log-normal size distribution is retrieved from measurements of spectral AOD, and analyzed the effects of possible errors due to uncertainties in an a priori assumed refractive index and sphericity parameter [26]. Román et al. (2018) used the GRASP algorithm to investigate the potential of retrieving aerosol properties from combined photometer and ceilometer data [27].

More recently, Lopatin et al. (2021) evaluated the limitations and capabilities of the multi-temporal retrievals of combined AERONET, Micro-pulse Lidar (MPL), and Compact Optical Backscatter Aerosol Detector (COBALD) observations using the GRASP algorithm [23]. Herrera et al. (2022) described the generation of the dynamic error estimates of GRASP retrieval and evaluated the reliability of the error estimations in the presence of random and systematic uncertainties for aerosol retrievals from observations of sun-sky radiometers alone and with lidar [28]. Herreras-Giralda et al. (2022) have extended the radiance calculations by GRASP SOS RT (Successive Orders of Scattering Radiative Transfer) code to the thermal-infrared electromagnetic spectrum, which allows the use in the extended spectral range [29].

None of these studies deals with the polarization information provided by the new AERONET polarized photometer in synergy with a three-wavelength elastic lidar.

In the current study, we use GRASP algorithm to analyze the potential of the retrieval of aerosol properties from synergy of data from AERONET sun-sky-lunar photometer with polarimetric capabilities with a three-wavelengths elastic lidar. The work is organized as follows: Section 2 presents the measurement site, the remote sensing instruments, a brief explanation of the degree of linear polarization, the description of the GRASP algorithm, and the methodology applied to this study, for example, the sensitivity tests; then, in Section 3, the noise-free and random noise inversions of the synthetic data with and without linear polarization are evaluated.

## 2. Materials and Methods

### 2.1. Characterization of the Measurement Site

Barcelona, Spain, is in a large coastal metropolitan area in the Mediterranean region (41.384°N, 2.120°E, 115 m a.s.l.), a region where climate is changing at a fast pace [30]. The measurement site is highly affected by urban and marine particles, Barcelona's local aerosols, and transported desert dust outbreaks and biomass burning from forest fires in the summer due to vegetation dryness [31]. Additionally, the Mediterranean region is also affected by desert dust outbreaks and European pollution episodes [32]. Morevoer, the atmosphere in Barcelona also has big contributions of pollen and spores [33,34].

Table 1 shows monthly averages of aerosol optical depth at 440 nm ($AOD_{440}$) and 440/870-nm Ångström Exponent ($AE_{440-870}$) measured by a sun photometer during 17 years of AERONET direct-sun observations at level 2 [35,36] in Barcelona. Over the whole period, the average $AOD_{440}$ is 0.18 and the average $AE_{440-870}$ is 1.31, i.e., small particles are predominant during the year. These results are typical of an urban-industrialized area, according to [37].

### 2.2. UPC Remote Sensing Instruments

The CommSensLab Optical Remote Sensing group at the Universitat Politècnica de Catalunya (UPC) operates several ground-based remote sensing instruments as part of the Aerosols, Clouds, and Trace Gases Research Infrastructure Network/European Aerosol Research Lidar Network/European Aerosol Research Lidar Network (ACTRIS/EARLINET) [38]. These instruments include a multi-wavelength high-power lidar system and an AERONET sun-sky-lunar photometer [39]. The instruments are located on the rooftop of the CommSensLab-UPC building "https://goo.gl/maps/nubSGYguMssbkZX99 (accessed on 18 September 2023)", in

Barcelona, Spain. This section presents and describes both instruments and the dataset used in this study.

**Table 1.** Monthly average of aerosol optical depth at 440 nm ($AOD_{440}$) and Ångström Exponent from 440 to 870 nm ($AE_{440-870}$) from 2004 to 2021 at level 2 in Barcelona without polarization data. The standard deviations (std) are in parenthesis.

| Property | Winter | | | Spring | | | Summer | | | Autumn | | |
|---|---|---|---|---|---|---|---|---|---|---|---|---|
| | **December** | **January** | **February** | **March** | **April** | **May** | **June** | **July** | **August** | **September** | **October** | **November** |
| $AOD_{440}$ (std) | 0.10 (0.07) | 0.11 (0.08) | 0.16 (0.12) | 0.16 (0.12) | 0.20 (0.12) | 0.18 (0.09) | 0.23 (0.11) | 0.25 (0.11) | 0.23 (0.11) | 0.21 (0.12) | 0.16 (0.11) | 0.11 (0.08) |
| $AE_{440-870}$ (std) | 1.37 (0.34) | 1.35 (0.38) | 1.36 (0.36) | 1.27 (0.34) | 1.25 (0.38) | 1.28 (0.34) | 1.34 (0.39) | 1.30 (0.37) | 1.32 (0.36) | 1.34 (0.32) | 1.23 (0.35) | 1.33 (0.31) |

### 2.2.1. Polarized Sun-Sky-Lunar Multispectral Photometer

An AERONET standard sun-sky photometer (CE318) [39], manufactured by Cimel Electronique, was operated at the UPC from 2004 to July 2021, when it was replaced by a sun-sky-lunar polarized multispectral photometer (CE318-TP9) [40]. The new photometer provides measurements at eight wavelengths in the spectral range from 340 to 1640 nm. The 937 nm channel is used by the AERONET network to determine the water vapor content. It also provides polarization data [6,41], direct-sun measurements, angular sky radiances [39], direct-lunar measurements, and columnar water vapor content [42–44]. The CE318-TP9 has a filter wheel containing 3 sets of 3 polarizers oriented with 120° separation, operating in infrared, ultraviolet, and visible spectral ranges. More technical specifications about the CE318-TP9 are displayed in Table 2.

**Table 2.** Technical specifications of the sun-sky-lunar multispectral photometer (CE318-TP9).

| Technical Specifications | |
|---|---|
| Tracking precision | 0.003° |
| Sun-moon tracking accuracy | 0.01° |
| Field of view | 1.29° |
| Digital count precision | <0.1% |
| Detectors | Silicon photodiodes and InGaAs |
| Spectral range | 340–1020 nm (Silicon detector) 1020–1640 nm (InGaAs detector) |
| Environmental temperature range | −20 °C to 50 °C |
| Power supply | 110–240 V |

The sky radiances are measured at eight wavelengths (340, 380, 440, 500, 675, 870, 1020, and 1640 nm), with azimuth angles varying from 0° to 180° for the left and right almucantars. Assuming a homogeneous atmosphere, scanning for both left and right branches can help reduce observational uncertainties as the value of optical mass remains constant during the measurements [24]. The azimuth angles below 3.5° were discarded, according to [36]. The frequency of the sky scans depends on the cloud-free conditions, the time of day, and the air masses. About 14 direct sun measurements per hour, more than 10 standard almucantar sequences, and around 8 polarized almucantar sequences are performed per day.

The polarization data used in this work are level 1.5, the only one available for the polarization photometer, with the cloud screening criterion [35] from Version 3 [45,46]. The data are available at seven wavelengths (380, 440, 500, 675, 870, 1020, and 1640 nm) during daytime measurements in the azimuthal range from 25° to 160° for left and right almucantar. The almucantar scan was chosen, the principal plane scan presenting too many limitations—horizon obstructions, inexact optical mass computation, and maximum scattering angle is typically smaller than stated theoretically [24], and the hybrid scan is still under validation [47]).

AERONET algorithms do not use polarization data yet [5,17,36]. They retrieve the aerosol size distribution (SD), refractive index (RI), sphere fraction (SF), single scattering albedo (SSA), and asymmetry factor (AF).

### 2.2.2. UPC/EARLINET Lidar System

The first UPC/EARLINET lidar system, operative since 1996, had two elastic channels (532 nm and 1064 nm). Then, another elastic channel at 355 nm and a Nitrogen-Raman channel (607 nm) were added to the system [48]. However, the main upgrade was designed and implemented by Kumar et al. (2011), which included a wavelength separation unit that provided the lidar backscattered returns of the three elastic channels (355, 532, and 1064 nm) and three Vibro-Rotational Raman (VRR) channels (387 and 607 nm, for nitrogen excited at 355 and 532 nm, respectively, and 407 nm, for water vapor excited at 355 nm) [49]. A few years later, two cross-polarization channels at 532 and 355 nm were implemented by means of attached auxiliary telescopes without altering the original system [50,51]. In 2020, a Pure-Rotational Raman (PRR) channel at 354 nm was incorporated [52].

The last upgrade of the UPC/EARLINET lidar system added a PRR channel at 530 nm [53]. Thus, the current UPC/EARLINET lidar system provides backscattering at three wavelengths (elastic channels at 1064, 532, and 355 nm), extinction at 532 and 355 nm (pure-rotational Raman channels at 530 and 354 nm), depolarization ratio at 532 and 355 nm, and Water Vapor Mixing Ratio (by means of a vibro-rotational channel at 407 nm), making it a singular system within the EARLINET community.

More technical details about the laser, telescope, and acquisition unit of the UPC/EARLINET multi-wavelength lidar system are in Table 3.

**Table 3.** Technical specifications of the UPC/EARLINET multi-wavelength lidar system.

| Technical Specifications | |
| --- | --- |
| Laser | Innolas Spitlight 400 |
| Zenith angle | $0°$ |
| Elastic wavelengths | 355, 532, and 1064 nm |
| Energy per pulse | 103, 154, and 97 mJ |
| Divergence | 28 mrad |
| Repetition rate | 20 Hz |
| Pulse duration | 3.6 ns |
| System configuration | Mono-static, vertical, biaxial |
| Spatial resolution | 3.75 m |
| Telescope model | Celestron CGE 1400 |
| Telephoto lens for depolarization measurements | TAIR-3S for 355 nm and 532 nm |
| Focal length of the telescope | 3.91 m |
| Telescope aperture Ø | 0.35 m |
| Detector | APD and PMT |
| Voltage Responsivity | $1.9 \times 10^6$ (355 nm), $1.5 \times 10^6$ (532 nm), and $3.7 \times 10^5$ (1064 nm) [V W$^{-1}$] |
| Spectral bandwidth | 1 nm for all wavelengths |
| Acquisition | Licel transient recorder TR40-80—mixed 250 MHz PC + ADC 40 Msps/12 bit |

### 2.3. Degree of Linear Polarization

CIMEL implemented the first polarizers in the 870 nm channel [41] and then extended the polarizers to the other channels, from 340 to 1640 nm [6], in order to provide the degree of linear polarization (DoLP) at a variety of scan angles. The scattered light in the atmosphere contains unpolarized and polarized components, so it can be described in terms of Stokes' parameters:

$$\{I, Q, U, V\}^T = \{I_{nat}, 0, 0, 0\}^T + \left\{I_{pol}, Q, U, V\right\}^T \tag{1}$$

where I is the total radiance, Q and U the components of linear polarization and V is the magnitude of circular polarization [54,55]; the superscript T stands for matrix transposition, and subscripts nat/pol mean natural and polarized light, respectively. The condition for the light to be natural is $Q = U = V = 0$, while for a polarized light $I_{pol} = I - I_{nat}$, and for a partially polarized light, the relation among these parameters is an inequality, $I^2 > Q^2 + U^2 + V^2$ [56].

The degree of polarization of scattered light is the ratio of the intensity of the polarization component to the total intensity [2], and it can be extracted from Stokes' parameters presented before. Its value is equal to 0 for non-polarized components and equal to 1 for total polarization. The DoLP is calculated neglecting Stokes' circular parameter (its value ranges from $10^{-2}$ to $10^{-5}$) [57] according to the equation below:

$$\text{DoLP} = \frac{\sqrt{Q^2 + U^2}}{I} \tag{2}$$

The polarized photometers only have three linear polarized filters. Thus, this paper is restricted to I, Q, U-parameters. If I, Q, and U are computed as a function of the polarized radiances from the photometer's filters, called $P_1$, $P_2$, and $P_3$, respectively, the Stokes' parameters (S) are obtained directly by:

$$S = \begin{bmatrix} I \\ Q \\ U \end{bmatrix} = \begin{bmatrix} \frac{2}{3}(P_1 + P_2 + P_3) \\ \frac{2}{3}(2P_1 - P_2 - P_3) \\ \frac{2}{\sqrt{3}}(P_3 - P_2) \end{bmatrix} \tag{3}$$

in which $P_1, P_2, P_3$ refer to the observed radiances in $\mu W\,cm^{-2}\,sr^{-1}\,nm^{-1}$ and the subscripts 1 to 3 refer to the orientations of the filters: 0° (1), 120° (2), and 240° (3). Considering the system of the Equation (3), Equation (2) can be rewritten in function of $P_1$, $P_2$, and $P_3$ as well:

$$\text{DoLP} = \frac{\sqrt{\left[\frac{2}{3}(2P_1 - P_2 - P_3)\right]^2 + \left[\frac{2}{\sqrt{3}}(P_3 - P_2)\right]^2}}{\frac{2}{3}(P_1 + P_2 + P_3)} \tag{4}$$

Solving Equation (4), the DoLP in function of $P_1$, $P_2$, and $P_3$ [6,7] is:

$$\text{DoLP} = \frac{2\sqrt{\left(P_1^2 + P_2^2 + P_3^2 - P_1P_2 - P_1P_3 - P_2P_3\right)}}{P_1 + P_2 + P_3} \tag{5}$$

Figure 1 is an example of DoLP from the aerosol scenarios in Barcelona, which are presented and discussed in the next sections. The coarse aerosol has dust contributions (Ångström Exponent < 1, for instance), while the fine aerosol is strongly influenced by urban aerosols (Ångström Exponent > 1, for instance). The DoLP takes values close to 0 at small scattering angles, and these values increase with increasing scattering angles. The measurements are not performed in the aureole area (angles < 25°) due to the photometer design and the sun saturation in this area.

### 2.4. GRASP Algorithm

GRASP code [21] has a two-module structure where the first one is the Numerical Inversion which is responsible for the retrievals through a statistical optimization fitting guided by the Multiterm Least Squares Method (LSM) to solve simultaneously a system of several independent equations [22], i.e., the best fit of the measurements gives the correct solution to the problem [7]. This method has previously been successful in other contexts [17,20,58,59]. The Numerical Inversion module of GRASP comprises two a priori constraints called single-pixel and multi-pixel inversions. The first one is a conventional approach in which remote sensing data is inverted completely independently [21]. The second one is also proposed by [21] and implements a simultaneous optimized inversion

of a large group of independent observations [22]. This approach improves retrieval consistency by using known limitations on the spatial and/or temporal variability of retrieved parameters [22]. The second module, called forward model, carries out the simulations in which the remote sensing observations are inverted to find iteratively the best inversion solution for the retrievals or to create synthetic data.

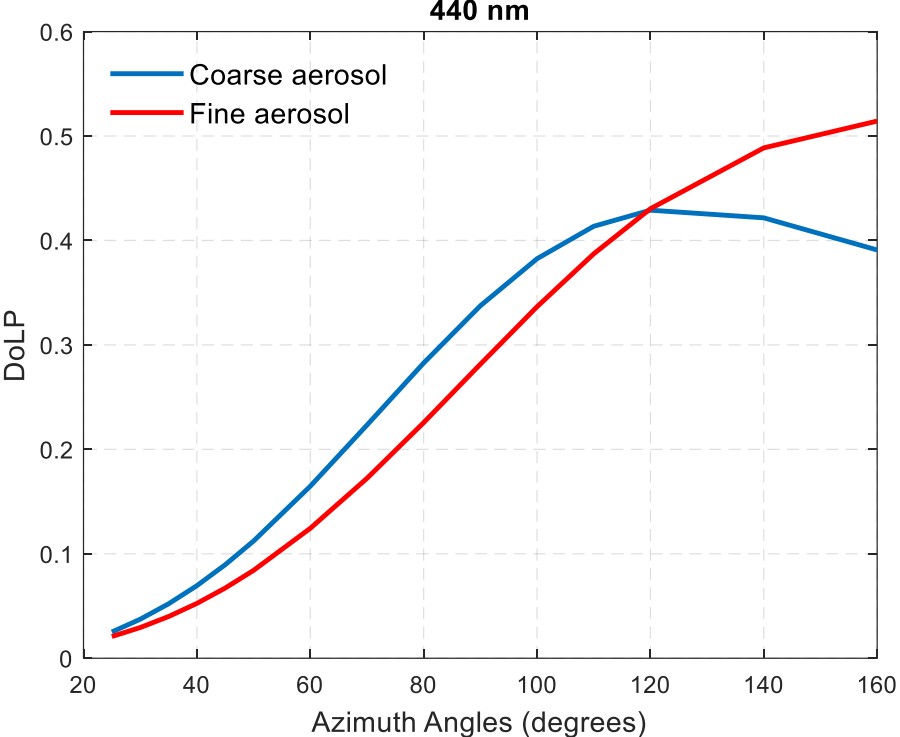

**Figure 1.** The DoLP at 440 nm from two aerosol scenarios with dominant coarse mode (blue) and dominant fine mode (red) in Barcelona, Spain.

The algorithm inputs are separated into two groups or files. The first group contains the observed data, while the second group contains all the retrieval settings information. Additionally, GRASP products can be classified as direct and derived, in which the amount of the outputs depends on the set of wavelengths, some configurations in the settings files, and if the properties are set up as bimodal or monomodal aerosols, for instance, most products from a retrieval between lidar systems and photometers are separated in coarse and fine mode [15]. The main aerosol-related products are summarized in Table 4. More details about their calculations are available in "https://www.grasp-open.com/doc/docgrasp.pdf (accessed on 18 September 2023)".

The data synergy was run by the GRASP algorithm with some combinations between the UPC/EARLINET lidar system and the new polarized sun-sky-lunar multispectral photometer. Table 5 presents the possible combinations between the UPC instruments, from a simple combination (D0), only with standard photometer data, to the most complex one (D1P+L), formed by AOD, radiances, and DoLP at seven wavelengths from the photometer plus RCS from the lidar system at three fundamental wavelengths (355, 532, and 1064 nm). The bold-print highlighted combinations were performed in this paper, being D0P+L, D1+L, and D1P+L studied for the first time.

**Table 4.** Main direct and derived aerosol-related products from the GRASP algorithm. These properties can be retrieved for fine and coarse modes.

| Outputs | Aerosol-Related Products | Acronym |
|---|---|---|
| Direct | Vertical distribution of the aerosol concentration | AVP (h) |
| | Size distribution | SD |
| | Real refractive index spectral | RRI (λ) |
| | Imaginary refractive index spectral distribution | IRI (λ) |
| | Ångström exponent spectral distribution | AE (λ) |
| | Aerosol optical depth spectral distribution | AOD (λ) |
| | Single scattering albedo spectral distribution | SSA (λ) |
| | Absorption AOD | AAOD (λ) |
| | Lidar ratio spectral distribution | LR (λ) |
| | Sphere fraction | SF |
| | Effective radii total | EffR |
| | Column-integrated volume concentration | VC |
| | Volume mean radius total | VMR |
| | Net forcing | NF |
| Derived | Ångström exponent profile | AEP (λ, h) |
| | Backscatter profile | β (λ, h) |
| | Extinction profile | α (λ, h) |
| | Single scattering albedo profile | SSAP (λ, h) |
| | Lidar ratio Profile | LRP (λ, h) |
| | Aerosol absorption profile | AAP (λ, h) |

**Table 5.** Daytime combinations between the UPC/EARLINET lidar system and the new sun-sky-lunar multispectral polarized photometer during the daytime (D). The standard photometer with 4λ is notated with 0, and the new one with 7λ is notated with 1. R refers to the radiances, P refers to the polarized data, and L refers to the lidar signals at 3λ. RCS refers to the range-corrected (lidar) signal.

| Notation | Inputs |
|---|---|
| **D0 (AERONET-like)** | AOD + R (4λ) |
| D0P | AOD + R + DoLP (4λ) |
| **D0+L** | AOD + R (4λ) + 3 RCS |
| **D0P+L** | AOD + R + DoLP (4λ) + 3 RCS |
| D1 | AOD + R (7λ) |
| D1P | AOD + R + DoLP (7λ) |
| **D1+L** | AOD + R (7λ) + 3 RCS |
| **D1P+L** | R + DoLP (7λ) + 3 RCS |

AERONET inversions with less than 3% sky error were selected. To guarantee the coincidence of the observations of both instruments, the lidar measurements were averaged over 60 min centered on the time of the AERONET inversions. Assuming a homogeneous and continuous atmosphere, the AOD and sky/polarized radiances were chosen in the closest possible time period, i.e., a ±16 min interval centered on almucantar inversion time, the same interval chosen by [47].

The GRASP algorithm requires some data pre-processing in which the polarized data and sky radiances are reduced and normalized, and then the symmetry between almucantar scan right and left branches is checked, eliminating those with differences greater than 20% (Figure 1 was plotted after the symmetry check). Background subtraction, range, trigger delay, overlap, dead-time corrections, and analog and photon-counting signal-gluing were applied to the lidar signals by the ACTRIS/EARLINET Single Calculus Chain (SCC) [60–62]. Furthermore, lidar profiles were normalized and re-sampled at 60 logarithmically spaced points, as recommended by Lopatin et al. (2013) [15].

### 2.5. Data Simulations of UPC Remote Sensing Instruments

This section reports the methodology for the sensitivity tests for D0P+L, D1+L, and D1P+L combinations in the GRASP algorithm to retrieve aerosol properties when DoLP and new channels of the photometer are added to the data. It is noteworthy that the GRASP algorithm can simulate synthetic data, activating the forward model in the settings file. In this mode, multiple initial guesses, SD, RRI, IRI, SF, and AVP, need to be set to run the code.

In this work, the simulations were performed for three aerosol scenarios in Barcelona:

- Scenario I: dominant coarse mode with high aerosol load ($AOD_{440}$ = 0.64),
- Scenario II: dominant coarse mode with low aerosol load ($AOD_{440}$ = 0.26),
- Scenario III: dominant fine mode with high aerosol load ($AOD_{440}$ = 0.33).

In all scenarios, the radiance uncertainties are less than or equal to 3% of the sky error uncertainties. Even though the lidar contributions already provide maximum benefits in synergy retrieval for retrievals with low aerosol loading [15], it is a better option to use AERONET retrievals with a load greater than 0.3 [24,63], that is, Scenarios I and III. As recommended by [26] and confirmed by [47], the lower limit of $AOD_{440}$ is 0.2 for the bimodal log-normal size distribution parameters in order to assure the quality retrievals of aerosol particles, since the uncertainty in the parameters dramatically increases as the aerosol load decreases [26,28]. So, for this study, a high load is defined as greater than 0.3.

Figure 2 describes the methodology for the synthetic data and the retrievals with and without noise. In the first step, the synthetic data was generated from the GRASP forward model based on previous data and initial guesses from some real observations of aerosols with dominant fine and coarse modes. The synthetic data contains new radiations, DoLP, AOD, and RCS for each aerosol scenario. Once the synthetic data were created, the next step was to run GRASP in inversion mode with a new settings file containing constant initial guesses for each aerosol model parameter (SD, RRI, IRI, SF, and AVP). This last step was to run two series of tests: (i) in noise-free conditions (NF), to test the stability of the GRASP inversions to minor noise [7], i.e., neither systematic nor random errors were introduced in this approach, and (ii) to add random noises (RN) to estimate the sensitivity [15] of GRASP inversions for the combinations of instruments in Table 5. Each series of tests were carried out for D0, D0+L, D0P+L, D1+L, and D1P+L combinations with a sample size of 100 per combination, adding random noises (1500 retrievals in total) and 1 noise-free retrieval for each combination (15 retrievals in total). Finally, the results of inversions without and with noises were compared with the reference values (simulated data). To control the NF and RN conditions, the GRASP standard deviations and the error type for each radiation/aerosol parameter were set to define the covariance matrix. More details about the covariance matrix can be found in [17,21,22].

### 2.6. Initial Guesses

Initial guesses for SD, RRI, IRI, SF, and AVP (see Table 4) were defined to create the synthetic simulated data by forward model (Figure 2). The simulated data was created to be as close as possible to atmospheric conditions in Barcelona, Spain. The values of these atmospheric parameters were obtained from AERONET network data and from previous studies such as [15,19,24,26]. The first initial guess is the SD (obtained from AERONET retrievals), a bimodal lognormal function corresponding to a mixture of aerosols with 22 log-spaced triangle bins, in which the first 10 bins (0.05–0.576 μm) correspond to the fine mode while the last 15 bins (0.335–15 μm) were assigned for the coarse mode, i.e., three bins overlap. The RRI and IRI were adapted from the aerosol models reported by [19]; data from Mexico City and Bahrain were used for the fine and coarse modes, respectively. The missing values of RRI and IRI from the new photometer and lidar system wavelengths (380, 500, 532, 1064, and 1640 nm) were interpolated or extrapolated [15,26], assuming that both components of the refractive index have a very weak spectral dependence [19,24,26].

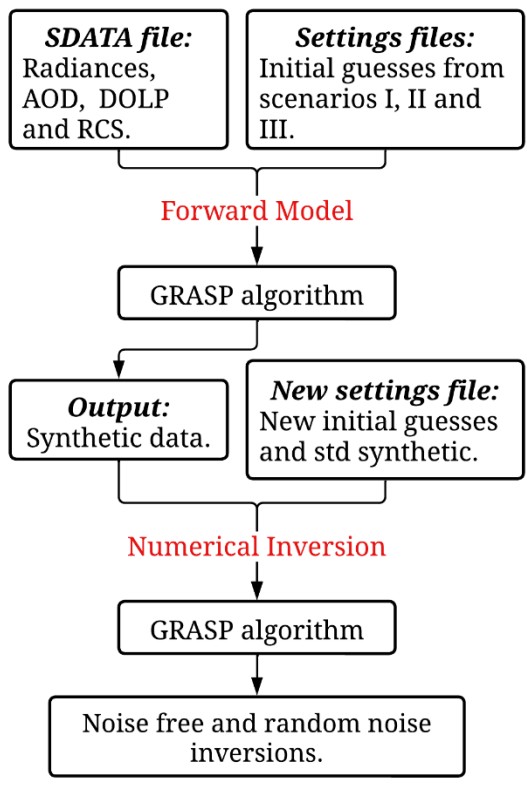

**Figure 2.** Methodology flowchart for the sensitivity study.

The values of SF [5] were obtained from AERONET inversions of Barcelona data. Concerning the aerosol shape, the GRASP algorithm considers each aerosol component as a mixture of polydisperse spheres and spheroids [15]. The last initial guess is the vertical distribution of the aerosol concentration (AVP), which describes the variability of aerosols at all altitudes from ground to space, being normalized to unity [15]. The AVP for fine and coarse modes was obtained following the approach from [15,28], in which the aerosol modes were represented in two different layers: an exponential distribution from the ground to maximum height (5 km) for fine mode and a Gaussian distribution for coarse mode, where the height of the layer was placed according to the aerosol layer from the local lidar signal. The vertical concentrations were multiplied by the fine-mode fraction at 500 nm from the Spectral Deconvolution Algorithm developed by [64], part of the AERONET processing algorithms. Finally, the AVP was normalized to unity according to Equation (16a) in [21] and Equation (8) in [15]. Furthermore, the characteristics called Surface Land BRDF Ross-Li [65–67] and Surface Land Polarized Maignan-Breon [68,69], were not included as inputs in this study, although they should be considered in research with retrievals of realistic measurements.

### 3. Results and Discussion

*3.1. Inversions of the Noise-Free Data*

The noise-free run for each combination considered in this study (Section 2.4) shows the best inversion convergence of the synthetic data (the first part of the flowchart in Figure 2). Table 6 shows the error types and the standard deviations used to retrieve aerosol properties from simulated data for each aerosol scenario and the residual after the last iteration of each combination (GRASP output), which should be less than 0.05 for good convergence [21]. For Scenario I, the convergence improves when the DoLP data is included (D0P+L and D1P+L), with a minimum for D1P+L after the last iteration of 0.00080. The standard deviation of the normalized radiances for Scenario II was set at 5% to achieve the smallest residual equal to 0.00055 for D0P+L and D1P+L. Note that the smallest residual (0.00043) is observed when more photometer channels are added to the

combination (D1+L). The last aerosol scenario has the best convergence for the D0P+L combination (residual equals 0.00111), but the weights for DoLP and lidar signal had to be increased (0.05 and 25%, respectively).

**Table 6.** Best noise-free convergences (Residual) from the simulated data for each combination (Comb.) and scenarios (Sc.). The normalized radiances (R), aerosol optical depth (AOD), degree of linear polarization (DoLP), Lidar Signal (LS), and their error types are in parentheses.

| Scenario | Comb. | R (rel) | AOD (abs) | DoLP (abs) | LS (rel) | Residual |
|---|---|---|---|---|---|---|
| I (High AOD and dominant coarse mode) | D0 | 3% | 0.01 | - | 20% | 0.04520 |
|  | D0+L |  |  | - |  | 0.00435 |
|  | D0P+L |  |  | 0.01 |  | 0.00183 |
|  | D1+L |  |  | - |  | 0.00593 |
|  | D1P+L |  |  | 0.01 |  | 0.00080 |
| II (Low AOD and dominant coarse mode) | D0 | 5% | 0.01 | - | 20% | 0.05149 |
|  | D0+L |  |  | - |  | 0.00753 |
|  | D0P+L |  |  | 0.01 |  | 0.00055 |
|  | D1+L |  |  | - |  | 0.00043 |
|  | D1P+L |  |  | 0.01 |  | 0.00055 |
| III (High AOD and dominant fine mode) | D0 | 3% | 0.01 | - | 25% | 0.02581 |
|  | D0+L |  |  | - |  | 0.00213 |
|  | D0P+L |  |  | 0.05 |  | 0.00111 |
|  | D1+L |  |  | - |  | 0.00148 |
|  | D1P+L |  |  | 0.05 |  | 0.00121 |

*3.2. Inversions with Random Noise Values*

This section mostly focuses on the RN retrievals, comparing the combinations that include DoLP (D0P+L and D1P+L) with those that do not (D0+L and D1+L) for the three aerosol scenarios. In this mode, a perturbation with a Gaussian distribution is added to the simulated data. The perturbation values were selected according to previous studies: 0.005 for AOD [15,39,70], 5% for radiances [27,28], 0.2 for 355 nm, 0.15 for 532 nm, and 0.1 for 1064 nm [15,28]. A value of 1% noise was used for DoLP [6,7,21]. Each aerosol scenario is presented by some descriptive statistics calculated from the RN inversions: the average represents the best-estimated retrieval, and the standard deviation represents the uncertainty around that retrieval.

3.2.1. Scenario I

Scenario I is characterized by a dominant coarse mode regime with $AOD_{440}$ = 0.64 and $AE_{440-870}$ = 0.29. Figures 3–5 show the microphysical and optical properties retrieved by the GRASP algorithm as well as the comparison with the reference (aerosol model), the NF and the RN inversions for D0, D0+L, D0P+L, D1+L, and D1P+L combinations. As it is seen in the figures, all the NF inversions have the best fittings with the reference when the DoLP is added into the D0P+L and D1P+L combinations for both aerosol modes. This is confirmed by the residuals in Table 6.

Figure 3 displays SD, RRI, and IRI retrieved by NF and RN inversions. The photometer-only combination (D0) is presented in all cases because it is the basic inversion that the AERONET network uses only angular direct and diffuse radiances and aerosol optical depth to derive the microphysical and optical properties of the aerosols [21,26]. It is important to emphasize that the D0 inversions compared to the ones from other combinations show some differences due to D0 being inverted as a one-component aerosol, while the other combinations were inverted with contributions from fine and coarse particles, as discussed in [15]. Their approach provides two independent vertical profiles of aerosol concentrations.

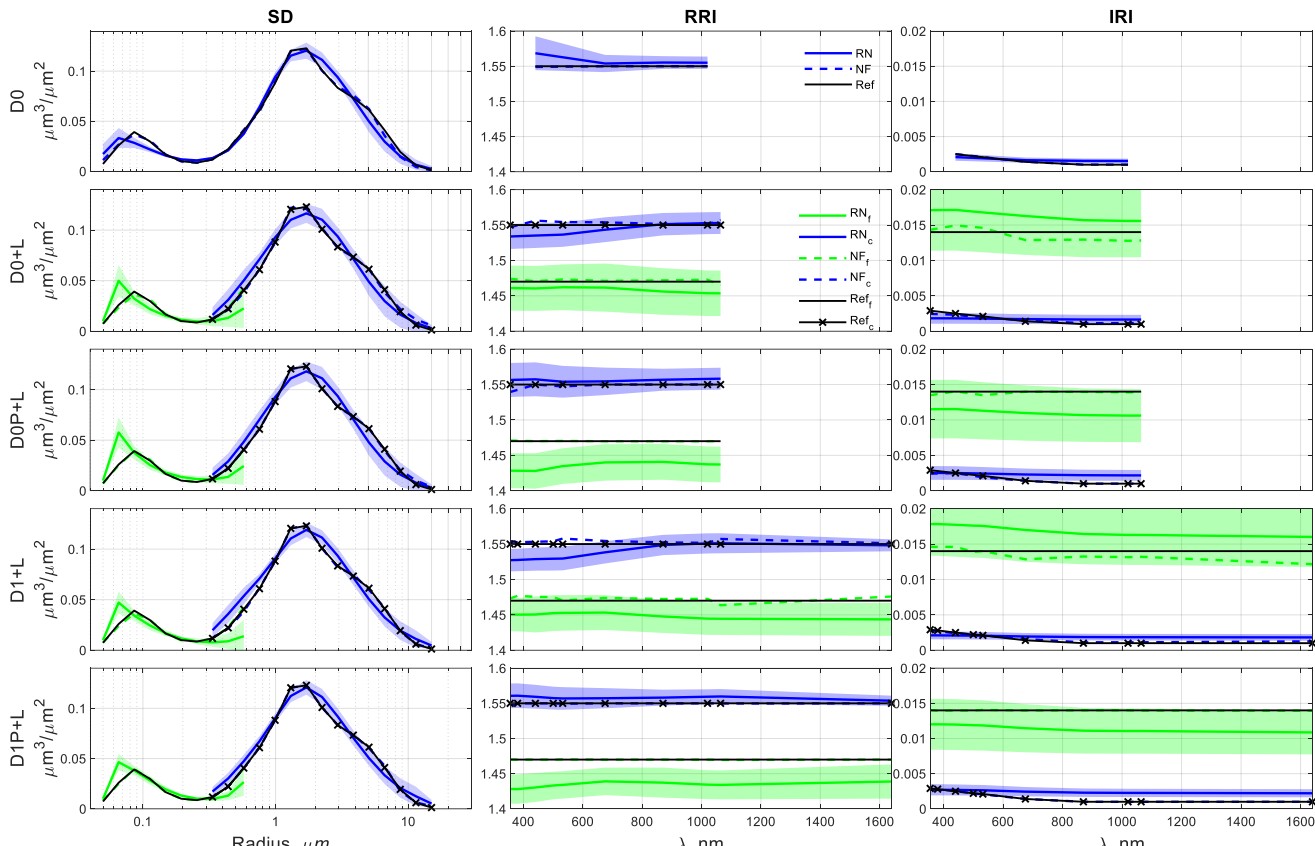

**Figure 3.** Microphysical properties of the aerosols retrieved from the simulated synergy between the polarized sun-sky-lunar multispectral photometer and the UPC/EARLINET lidar system. The columns from left to right refer to the SD, RRI, and IRI, while the rows from top to bottom refer to the combinations D0, D0+L, D0P+L, D1+L, and D1P+L. The green color refers to the fine mode (subscript f), and the blue color refers to the coarse mode (subscript c) or one-component aerosol in the case of D0. Colorful solid lines are the averaged profiles of the random noise (RN) retrievals, shaded areas are the standard deviation of RN retrievals, dashed lines are the noise-free (NF) retrievals, and solid black lines with and without stars are the reference (Ref).

The uncertainty in the inverted values for coarse-mode SD (first column of Figure 3) shows an increase of 2% and 3% when DoLP is included in the synergy (D0+L and D1+L compared to D0P+L and D1P+L, correspondingly), followed by a slight decrease in the underestimation of the reference (correlation coefficient values vary from 0.9794 to 0.9803 for D0+L and D0P+L and from 0.9790 to 0.9865 for D1+L and D1P+L). This improvement was not found in [7], as lidar channels were absent from their research. Nevertheless, the averaged RN inversions overestimate the fine mode as compared with the reference and shift the peak of this mode to radius 0.066 μm in all the combinations, which could be attributed to (i) the lidar contributions at the shorter edge of the SD and (ii) the GRASP algorithm adjusting the dominant mode better in the retrieved properties [15]. The shifted peak is also observed in the next scenario. The fine-mode overestimations are expected due to the low AOD in this mode that decreases to values under 0.05 at longer wavelengths, as the GRASP algorithm tends to fit better with the dominant mode [15].

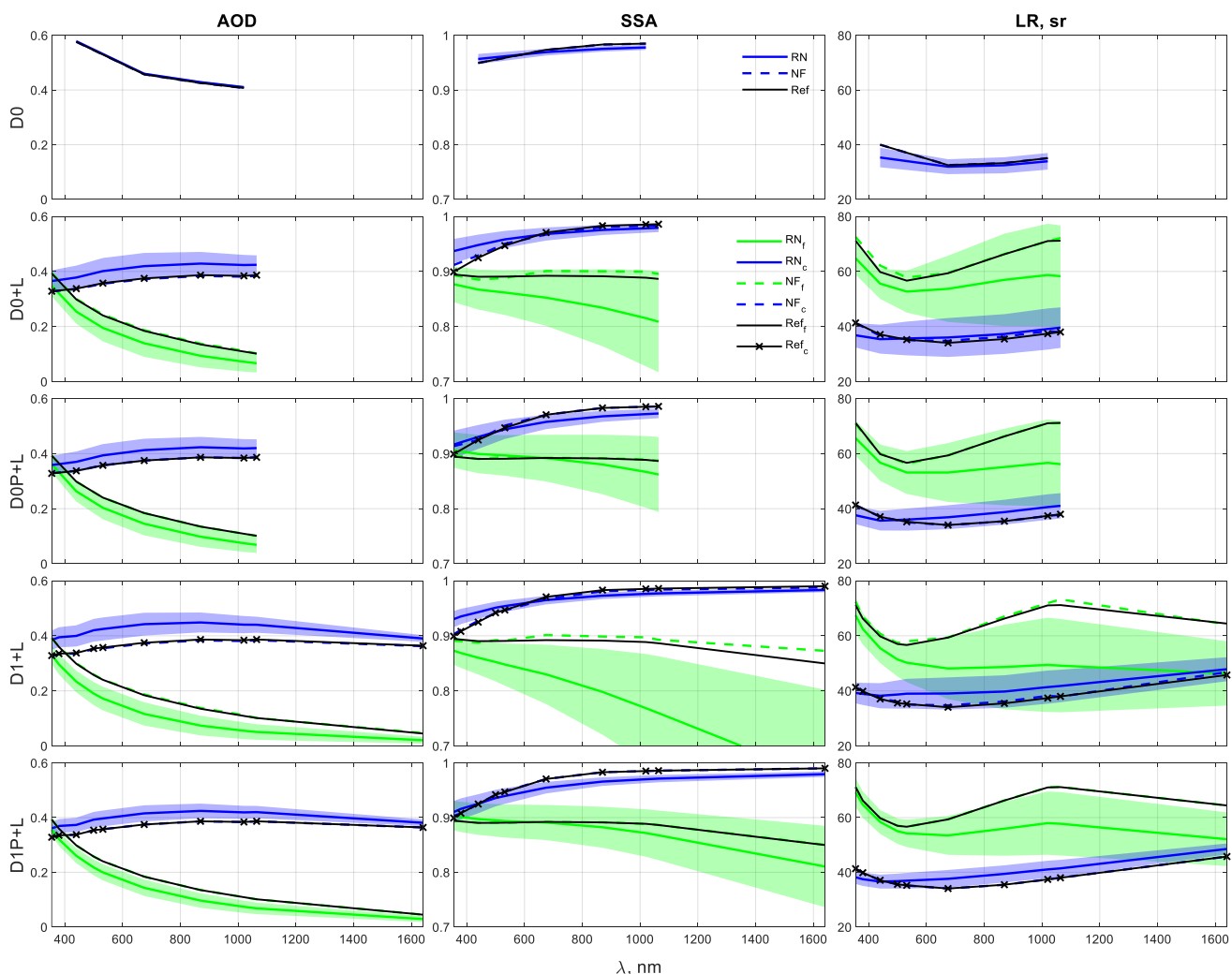

**Figure 4.** Optical properties of the aerosols retrieved from the simulated synergy between the polarized sun-sky-lunar multispectral photometer and the UPC/EARLINET lidar system. The columns from left to right refer to the AOD, SSA, and LR, while the rows from top to bottom refer to the combinations D0, D0+L, D0P+L, D1+L, and D1P+L. The green color refers to the fine mode (subscript f), and the blue color refers to the coarse mode (subscript c) or one-component aerosol in the case of D0. Colorful solid lines are the averaged profiles of the random noise (RN) retrievals, shaded areas are the standard deviation of RN retrievals, dashed lines are the noise-free (NF) retrievals, and solid black lines with and without stars are the reference (Ref).

The GRASP algorithm improves the estimations of the coarse-mode RRI at longer wavelengths while underestimating them for shorter wavelengths, as shown in the second column of Figure 3. For instance, the uncertainty values show a decrease of 21% at 870 nm, 26% at 1020 and 1064 nm, and 15% at 1640 nm when comparing D1+L and D1P+L. These RRI uncertainty reductions can be explained by the influence on the linear polarization of big particles, which is more noticeable at larger wavelengths for a dominant coarse-mode aerosol. The improvements with DoLP in the estimation of the coarse-mode RRI at shorter wavelengths were also observed by [7]. Furthermore, the uncertainties in the fine-mode RRI estimation decrease when D0P+L, D1+L, and D1P+L combinations are used, achieving a reduction of 24% when comparing D0+L with D0P+L and 23% for D1P+L with respect to D1+L.

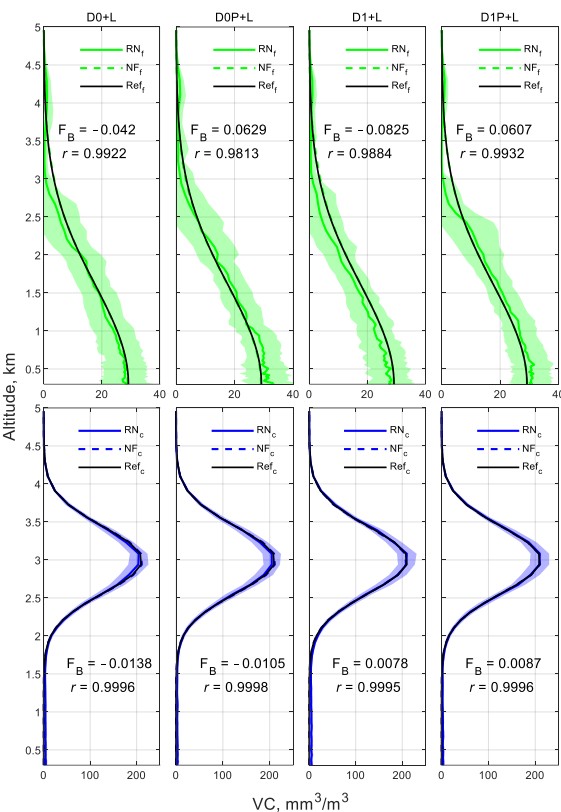

**Figure 5.** Volume Concentration (VC) retrieved from the simulated synergy between the polarized sun-sky-lunar multispectral photometer and the UPC/EARLINET lidar system. The columns from left to right refer to the combinations D0+L, D0P+L, D1+L, and D1P+L. The green color refers to the fine mode (subscript f), and the blue color refers to the coarse mode (subscript c). Colorful solid lines are the averaged profiles of the random noise (RN) retrievals, shaded areas are the standard deviation of RN retrievals, dashed lines are the noise-free (NF) retrievals, and solid black lines with and without stars are the reference (Ref).

The third column of Figure 3 shows the IRI, in which the coarse mode of the reference is slightly overestimated with an increase in uncertainties at longer wavelengths, followed by a better adjustment at shorter wavelengths when DoLP is an input of GRASP algorithm. Li et al. (2009) attributed the small change in the coarse-mode estimation to the low DoLP contribution from non-spherical particles [6]. Although the estimation of the fine-mode IRI has uncertainty reductions of 29% (D0+L compared with D0P+L) and 26% (D1+L compared with D1P+L), there is no improvement in retrieval accuracy when including the DoLP in the data, as shown in the third column of Figure 3.

The first column of Figure 4 displays the AOD spectrum derived with and without DoLP. The fine- and coarse-mode AOD have small changes when the linear polarization is considered in synergy, improving the estimation of both modes, especially when comparing the D1+L with the D1P+L combination. Fedarenka et al. (2016) also found a small change in the AOD for inversions with linear polarization [7].

The second column of Figure 4 displays the spectral SSA. Higher accuracies are achieved in the retrievals with DoLP for fine-mode SSA (73% and 83% of error reductions for D0+L/D0P+L and D1+L/D1P+L comparisons). However, the coarse-mode SSA only has improvements at shorter wavelengths, decreasing the underestimation of the reference. The fine mode has its uncertainties reduced up to 26% (D0+L/D0P+L comparison) and 56% (D1+L/D1P+L comparison) at visible and infrared channels.

The last column of Figure 4 presents the LR as part of GRASP-output comparisons with and without DoLP. The addition of DoLP improves the accuracy in the retrievals, especially when comparing D1+L with D1P+L combinations for both modes (40% and 13%

of error reductions for fine and coarse mode, respectively), but the averaged RN inversions slightly overestimate the coarse-mode LR. This overestimation is also present in the D1+L combination (without DoLP). The maximum reductions in the uncertainties are 38% and 54% at longer wavelengths for comparisons between D0+L and D0P+L and between D1+L and D1P+L, respectively, in the coarse mode. The fine-mode LR also has evident changes in retrieval accuracy; for instance, the insertion of DoLP decreases the uncertainties by ~16% at the wavelengths, comparing D0+L with D0P+L, and a maximum reduction of 36% at longer ones, comparing D1+L with D1P+L. This similar tendency was observed in the previous aerosol properties. In general, the results of the optical properties from Scenario I agree well with [15,24,63], who observed adequate retrievals of aerosol optical parameters for cases with high optical thickness.

To better quantify the contribution of the DoLP to the volume concentration (VC(h)), two statistical indicators were calculated. The first one is the correlation coefficient (r) to quantify the intensity of the linear relationship between the reference and averaged RN inversions, that is, the capability of averaged RN inversions to reproduce the shape of the aerosol volume concentration profiles. The second statistical indicator is the fractional bias ($F_B$) to express the underestimation or overestimation caused by only the systematic errors [71] between the reference and averaged RN inversions. The $F_B$ was computed by the following equation in [72]:

$$F_B = \frac{2}{n} \left( \sum_1^n \frac{(M_{ref} - O_{RN})}{(M_{ref} + O_{RN})} \right) \tag{6}$$

where n is the sample size, $M_{ref}$ is the reference, and $O_{RN}$ is the averaged profile of the RN inversions. $F_B = 0$ indicates perfect overlapping between the profiles, i.e., free from bias, $F_B = -2$ indicates an extreme underestimation, and $F_B = 2$ means an extreme overestimation. For this study, $F_B$ and r are only representative of the main aerosol layer from each profile to avoid a distorted estimation caused by too small values in the ratio (Equation (6)) or by another aerosol layer in the same profile.

The fine- and coarse-mode VC were computed by multiplying the AVP by the aerosol column-integrated volume concentration and they are displayed in Figure 5. In the fine mode, the RN inversions with DoLP (D0P+L and D1P+L) slightly overestimate the reference (negative $F_B$ values become positive ones) in the layer below 2 km, followed by a no significant change on r. The coarse-mode VC does not have a significant change in r as well; however, the DoLP seems to slightly decrease the underestimation of the reference (reduction in $F_B$ value), comparing D0+L with D0P+L. The slight overestimation and underestimation of the fine and coarse modes, respectively, were pointed out by [15] for retrievals without DoLP, making these VC estimations expected. The small contribution of the DoLP in the concentration profiles can be related to the small differences between the aerosol loads for each combination (first column of Figure 4), in which the aerosol load is taken into consideration when estimating the AVP according to Equation (16a) in [21] and Equation (8) in [15].

Thus, in contrast with previous literature [7], who did not observe any advantages for adding DoLP in their retrievals of SD, IRI, and SSA, this study shows significant advantages in the retrieval of the dominant coarse-mode aerosol for those aerosol properties plus SF, RRI, and LR. Nevertheless, we emphasize that the DoLP has the smallest contribution to the retrievals of SD, AOD, and VC. The observed improvements can also be related to the contribution from the lidar system in the synergy between the polarized sun-sky-lunar multispectral photometer and the UPC/EARLINET lidar system.

### 3.2.2. Scenario II

The second aerosol scenario is a dominant coarse mode with low aerosol loading ($AOD_{440} = 0.26$) and $AE_{440-870}$ equal to 0.43. Figure 6 shows the microphysical properties (SD, RRI, and IRI) retrieved by NF and RN conditions for D0, D0+L, D0P+L, D1+L, and D1P+L combinations. The first column of Figure 6 shows the fine-mode SD that overestimates the reference with a shifted peak, which was also observed in the previous

aerosol scenario. The correlation between the averaged RN inversions and the reference in the coarse-mode SD slightly increases from 0.9939 to 0.9969 and from 0.9962 to 0.9971 (D0+L/D0P+L comparison and D1+L/D1P+L comparison, respectively). The increase in the correlation between D0+L and D0P+L is followed by an increase in the uncertainties (9%); on the other hand, the uncertainties are reduced by 53% when DoLP is added to the D1+L combination.

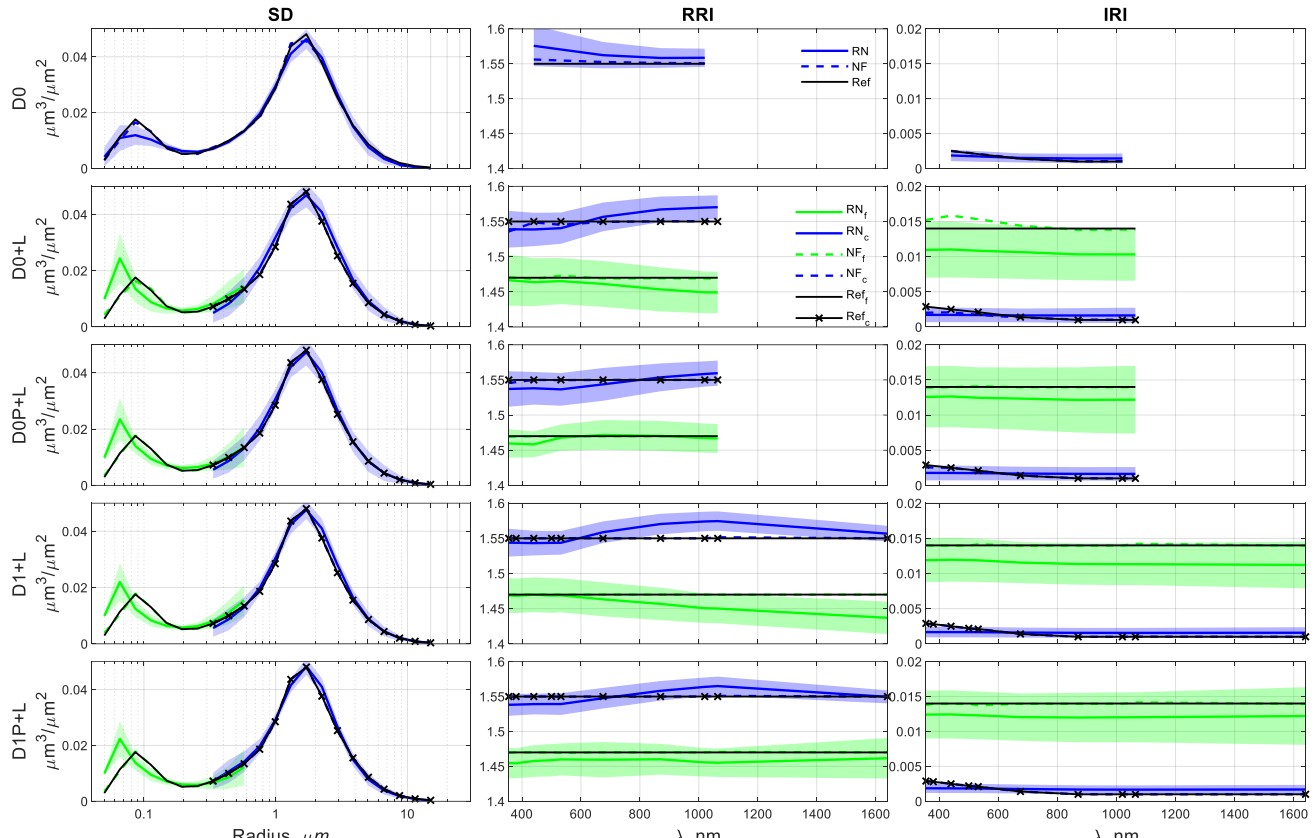

**Figure 6.** Idem as Figure 3 for Scenario II.

The RRI is depicted in the second column of Figure 6. The coarse mode decreases the overestimation of the reference at longer wavelengths, with a maximum reduction of 16% (1640 nm) in the uncertainties. At shorter wavelengths, the uncertainties are decreased up to ~20%, although the underestimation of the reference continues in these spectral regions when DoLP is added to the inversions. The fine mode decreases the underestimation of the reference at longer wavelengths with DoLP, but maximum uncertainty reductions (46% and 13%) only occur at shorter wavelengths (D0P+L and D1P+L, correspondingly).

In the case of the IRI, third column of Figure 6, the coarse mode seems to not have any improvements with DoLP; however, in the fine mode, DoLP only decreases the underestimations of the reference.

Figure 7 presents AOD, SSA, and LR as retrievals with NF and RN conditions compared to the reference. Both retrieved modes of AOD in the first column seem to adjust better to the reference with the DoLP in the GRASP inversions, mainly reducing the RN uncertainties up to 8% in the fine mode and 7% in the coarse mode at longer wavelengths. Low sensitivity has also been observed in the AOD by [7].

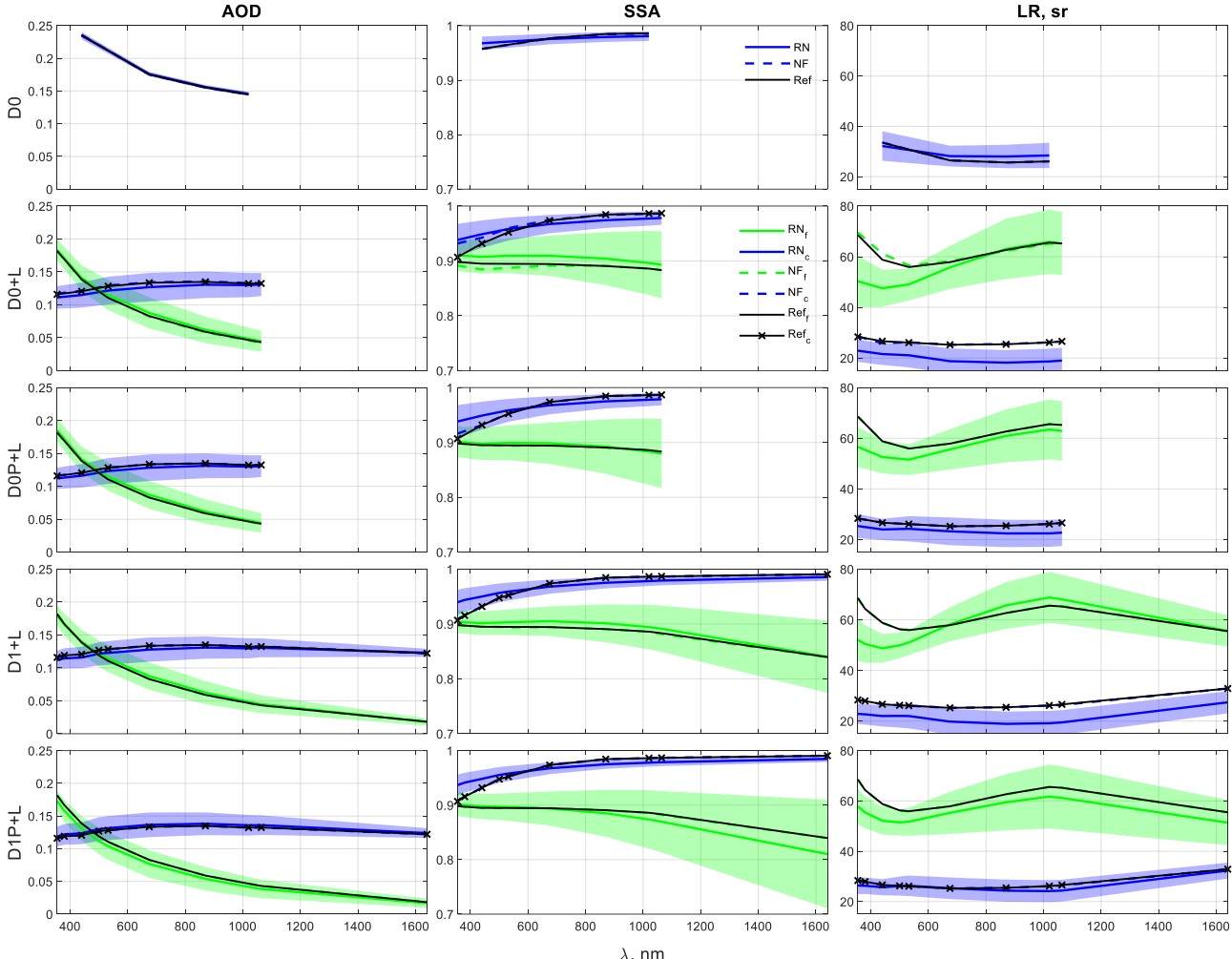

**Figure 7.** Idem as Figure 4 for Scenario II.

As in the previous aerosol scenario, SSA and LR have the most noticeable gains with DoLP in the GRASP retrievals. The coarse-mode SSA reduces the RN uncertainties up to 10% (D0+L/D0P+L comparison) and 18% (D1+L/D1P+L comparison) at longer wavelengths, but the averaged RN inversions remain underestimating the reference, as can be seen in the second column of Figure 7. The fine-mode SSA reduces the uncertainties by 2.5% at shorter wavelengths, followed by a decrease in the overestimation of the reference when DoLP is inserted in the D0+L combinations (error reductions of 77%).

The LR is shown in the third column of Figure 7. In the fine mode, there is a decrease in the underestimation of the reference at shorter wavelengths, which is observed in D0P+L and D1P+L with maximum uncertainty reductions of 22% and 16% at 355 nm, respectively. However, there is no enhancement at longer wavelengths when DoLP is considered in the inversions. For the coarse mode, the underestimation of the reference decreases for D0P+L and D1P+L combinations (error reductions of 54% and 76%, respectively), with a maximum uncertainty reduction of 28% at 1640 nm.

The simulated VC and its NF and RN inversions are plotted in Figure 8. The underestimation of the reference remains with DoLP (D0P+L and D1P+L combinations) in the fine mode, that is, no significant change in $F_B$ and r values. The use of DoLP seems to be sensitive to the mixing region between 2 and 3 km, overestimating the reference in this region. The coarse-mode VC decreases the overestimation of the reference (reduction in the $F_B$ value) when D1+L is compared with D1P+L. Finally, DoLP seems not to have made any important contributions to the RN inversions for both modes.

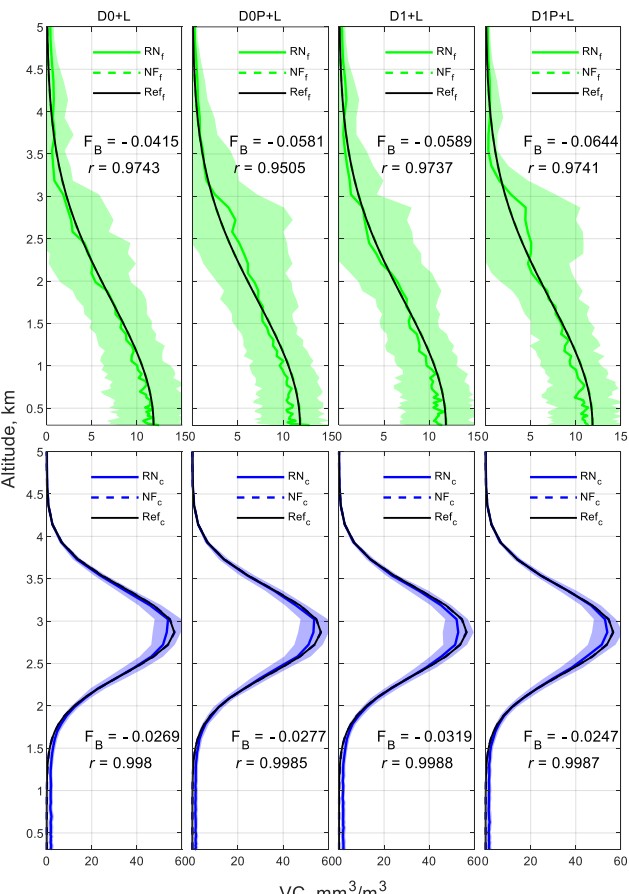

**Figure 8.** Idem as Figure 5 for Scenario II.

### 3.2.3. Scenario III

Scenario III is described as a dominant fine mode with high aerosol loading ($AOD_{440}$ = 0.33), $AE_{440–870}$ equal to 1.16, and a small contribution of coarse particles. The SD in the first column of Figure 9 does not show any improvements with DoLP to the inversions; however, there is a more accurate retrieval at the edges of the modes when lidar channels and more photometer channels are added to the synergy (comparison among D0, D0+L, and D1+L). The shift in the edge is expected at higher radii due to the use of lidar signals in the inversions, providing additional information at scattering angles of 180° [15,28,73].

The next property, RRI, is shown in the second column of Figure 9. Its reference is slightly overestimated in the fine mode with DoLP information in the retrievals, followed by uncertainty reductions up to 30% (D0+L/D0P+L comparison) and 23% (D1+L/D1P+L comparison) at shorter wavelengths. The averaged RN inversions slightly reduce the overestimation of the reference in the coarse mode for the D0P+L and D1P+L combinations, although the uncertainty reductions are observed only in the D1P+L combination, with maximum reduction at longer wavelengths (36% at 1640 nm).

The IRI in the third column of Figure 9 shows a decrease in the fine-mode underestimation of the reference for D0P+L and D1P+L combinations, followed by an uncertainty reduction of 16% at shorter wavelengths (D0P+L). The coarse-mode IRI has uncertainty reductions of around 30% and 10% at all wavelengths for the D0P+L and D1P+L combinations, respectively. The minor sensitivity to the addition of DoLP is accordant with the study of [7] for IRI in a dominant fine-mode aerosol. The refractive indexes' results for this aerosol scenario agree with [6], in which they explained that polarization is more sensitive to RRI than IRI for particles with AE > 0.5.

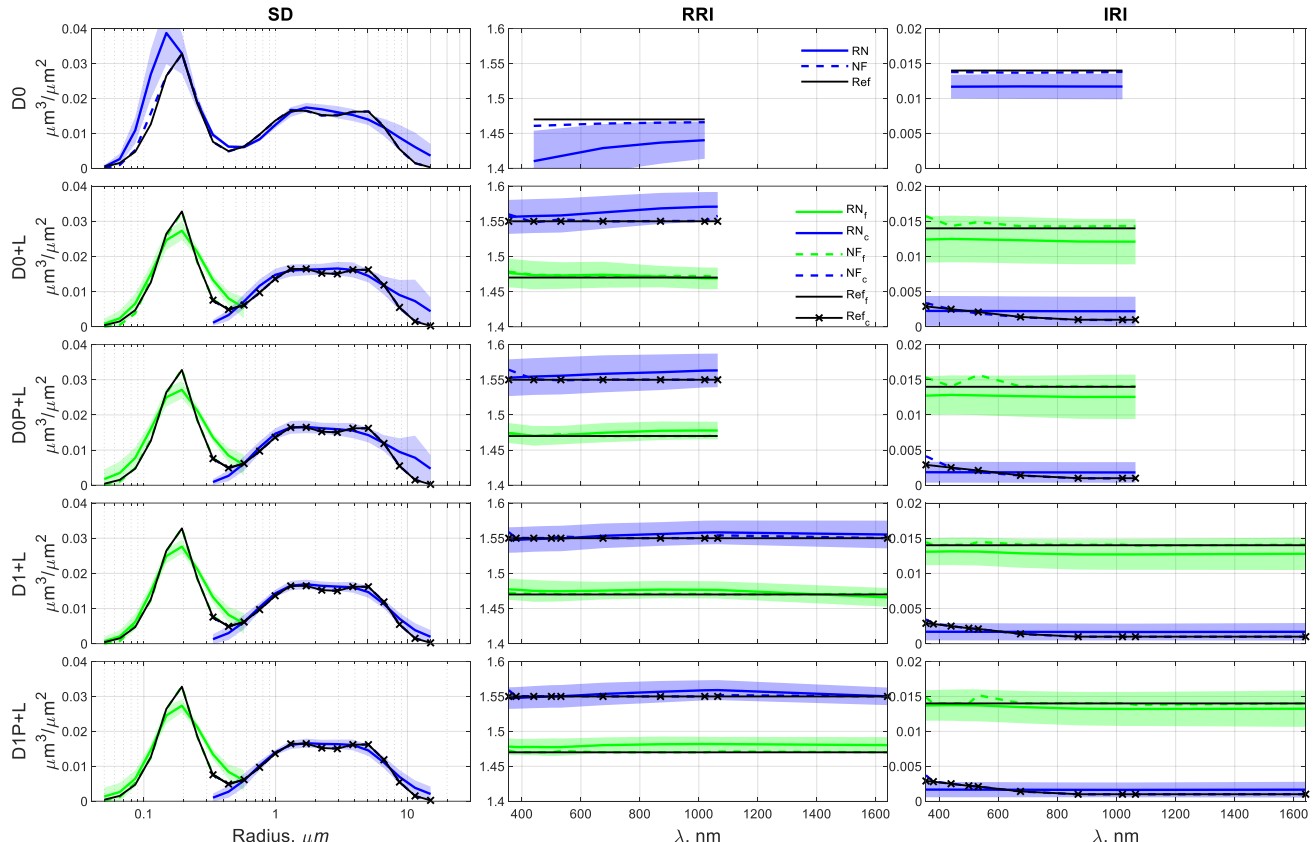

**Figure 9.** Idem as Figure 3 for Scenario III.

Figure 10 shows the optical properties retrieved by the GRASP algorithm. The retrieved AOD has no significant enhancement when DoLP is inserted in the retrievals for this aerosol scenario. The fine-mode SSA has a maximum uncertainty reduction of 15% at shorter wavelengths (D0+L/D0P+L comparison), and there is a slight decrease in the overestimation of the reference at shorter wavelengths (D0P+L and D1P+L). In the coarse-mode SSA, the DoLP also slightly decreases the underestimations of the reference, with a reduction in uncertainties of around 20% in all channels (D0+L/D0P+L comparison), while the maximum reduction of 6% is observed for the D1P+L combination at longer wavelengths.

The last property in the third column of Figure 10 is the LR, in which its fine mode has a slight decrease in the underestimation of the reference at shorter wavelengths when DoLP is added to the inversions. The uncertainty reduction for the D0P+L combination reaches a maximum of 18% at shorter wavelengths in this mode. The DoLP slightly decreases the underestimation of the reference for D0P+L and D1P+L combinations for coarse-mode LR, showing an uncertainty reduction up to 16% at 1640 nm for the D1P+L combination.

The fine and coarse-mode VC are shown in Figure 11. In the main layer (below 1 km) of the fine mode, the DoLP slightly decreases the underestimation of the reference (reductions in the $F_B$ value for D0P+L), followed by a small increase in the correlation (r) between the averaged RN inversions and the reference for D0P+L and D1P+L combinations. In the coarse-mode VC, the DoLP also slightly decreases the underestimation of the reference; however, it seems not to have improvements for the D1+L/D1P+L comparison. In general, the DoLP has minor improvements on the VC estimations for both modes in all aerosol scenarios.

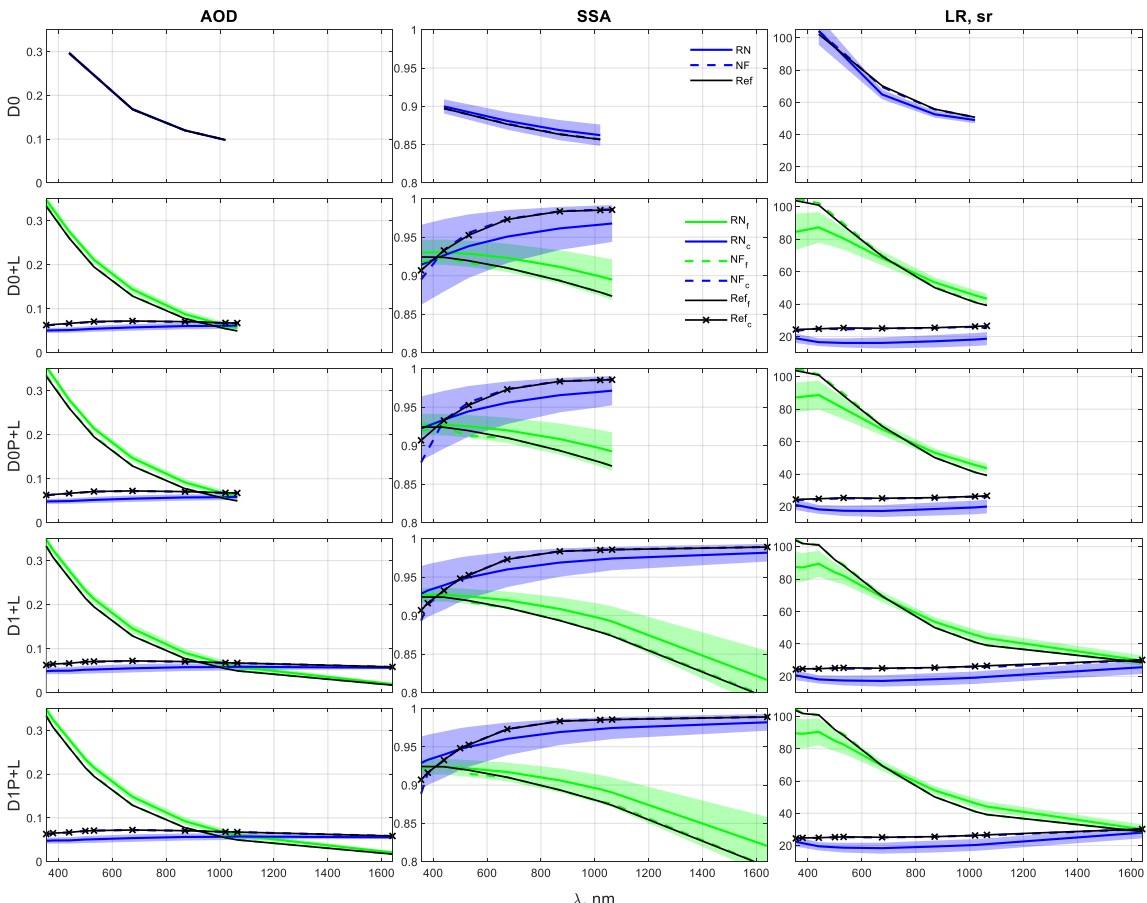

**Figure 10.** Idem as Figure 4 for Scenario III.

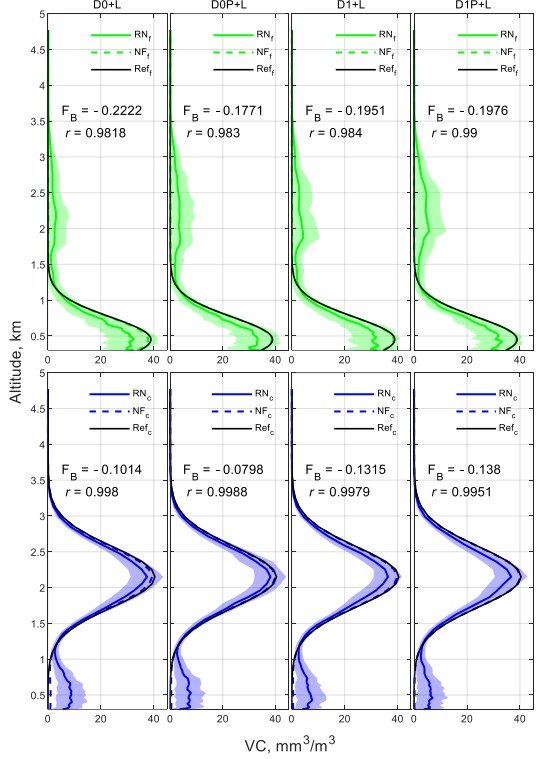

**Figure 11.** Idem as Figure 5 for Scenario III.

3.2.4. Root Mean Square Error and SF

In this sectionwe compute the root mean square error (RMSE), to assess the dispersion ofthe RN inversions. Furthermore, the last aerosol property (SF) is also discussed in this section (both are included in Table 7, which summarizes the results of the previous sections). The RMSE was calculated by the equation below:

$$\text{RMSE} = \sqrt{\frac{1}{n}\sum_{1}^{n}|M_{\text{ref}} - O_{\text{RN}}|^2} \tag{7}$$

where n is the sample size, $M_{\text{ref}}$ is the reference, and $O_{\text{RN}}$ is the averaged profile of the RN inversions. If RMSE values are equal to 0, there is an optimum fit between the averaged RN inversions and the reference, i.e., a correlation coefficient equal to 1. It is important to mention that RMSE is calculated for each aerosol mode, so there is no accurate information about how the RMSE increase or decrease is by channel in the GRASP retrievals.

**Table 7.** Root mean square error (RMSE) for fine and coarse modes of the size distribution (SD), real refractive index (RRI), imaginary refractive index (IRI), aerosol optical depth (AOD), single scattering albedo (SSA), lidar ratio (LR), and sphere fraction (SF) and its standard deviation (std) retrieved by the synergy between the polarized sun-sky-lunar multispectral photometer and the UPC/EARLINET lidar system for each combination (Comb.) and scenario (Sc.). The green color highlights the RMSE reductions when DoLP is added in the synergy (D0P+L and D1P+L combinations), and the orange color highlights the RMSE reductions for D1+L retrieval compared with D0+L.

| Sc. | Comb. | SD | | RRI | | IRI | | AOD | | SSA | | LR | | SF * | |
| --- | --- | --- | --- | --- | --- | --- | --- | --- | --- | --- | --- | --- | --- | --- | --- |
| | | Fine | Coarse | Fine | Coarse | Fine | Coarse | Fine | Coarse | Fine | Coarse | Fine | Coarse | Mean (Std) | Total |
| I | D0+L | 0.011 | 0.008 | 0.012 | 0.010 | 0.0024 | 0.0007 | 0.042 | 0.041 | 0.051 | 0.018 | 8.55 | 2.26 | 0.035 (0.055) | 0.023 |
| | D0P+L | 0.012 | 0.008 | 0.035 | 0.007 | 0.0030 | 0.0009 | 0.035 | 0.035 | 0.014 | 0.012 | 9.61 | 2.80 | 0.010 (0.022) | 0.001 |
| | D1+L | 0.012 | 0.009 | 0.021 | 0.016 | 0.0031 | 0.0007 | 0.060 | 0.059 | 0.096 | 0.016 | 13.54 | 3.28 | 0.024 (0.042) | 0.013 |
| | D1P+L | 0.009 | 0.007 | 0.037 | 0.009 | 0.0025 | 0.0009 | 0.036 | 0.035 | 0.016 | 0.012 | 8.13 | 2.85 | 0.004 (0.010) | 0.008 |
| II | D0+L | 0.005 | 0.002 | 0.014 | 0.015 | 0.0034 | 0.0007 | 0.003 | 0.005 | 0.013 | 0.015 | 8.55 | 6.42 | 0.360 (0.093) | 0.164 |
| | D0P+L | 0.005 | 0.001 | 0.006 | 0.010 | 0.0016 | 0.0007 | 0.003 | 0.004 | 0.003 | 0.015 | 5.60 | 2.96 | 0.285 (0.068) | 0.089 |
| | D1+L | 0.005 | 0.002 | 0.015 | 0.014 | 0.0024 | 0.0007 | 0.003 | 0.004 | 0.008 | 0.016 | 8.12 | 5.62 | 0.319 (0.076) | 0.123 |
| | D1P+L | 0.005 | 0.001 | 0.013 | 0.011 | 0.0018 | 0.0007 | 0.007 | 0.003 | 0.011 | 0.015 | 5.85 | 1.34 | 0.232 (0.036) | 0.036 |
| III | D0+L | 0.003 | 0.003 | 0.003 | 0.015 | 0.0017 | 0.0009 | 0.013 | 0.012 | 0.015 | 0.017 | 9.75 | 8.12 | 0.470 (0.176) | 0.183 |
| | D0P+L | 0.003 | 0.003 | 0.006 | 0.009 | 0.0013 | 0.0007 | 0.017 | 0.015 | 0.012 | 0.014 | 8.77 | 6.75 | 0.400 (0.106) | 0.113 |
| | D1+L | 0.003 | 0.002 | 0.006 | 0.005 | 0.0011 | 0.0008 | 0.014 | 0.014 | 0.013 | 0.012 | 8.78 | 6.53 | 0.427 (0.104) | 0.140 |
| | D1P+L | 0.003 | 0.002 | 0.010 | 0.005 | 0.0006 | 0.0008 | 0.016 | 0.016 | 0.012 | 0.012 | 7.86 | 5.33 | 0.366 (0.083) | 0.079 |

\* The RMSE of the SF is derived in one mode (total).

The RMSE reductions with DoLP are highlighted in light green, and those reductions due to the new photometer channels (D1+L) are highlighted in orange in Table 7 to be compared with D0+L values. The columns 3 and 4 in Table 7 show the RMSE of the SD. The averaged RN inversions fit the reference better and decrease the RMSE values only when DoLP is added to the D1+L combination for both SD modes in Scenario I (RMSE reductions of 25% for fine mode and 22% for coarse mode) and for coarse mode in Scenario II (RMSE reductions of 50% for D0P+L and D1P+L combinations).

Relating to the RRI (columns 5 and 6 in Table 7), the DoLP reduces the RMSE in the coarse mode in Scenario I (30% and 44% for D0P+L and D1P+L, respectively), both modes of Scenario II (57%/13% in the fine mode of D0P+L/D1P+L and 33%/21% in the coarse mode of D0P+L/D1P+L), and only the coarse mode of Scenario III (40% in D0P+L).

The IRI is shown in columns 7 and 8 in Table 7. The reductions in RMSE values are observed in the fine modes of all scenarios, except for D0P+L in Scenario I, with the largest one for the fine-mode IRI of Scenario II (53% comparing D0+L with D0P+L). However, the only enhancement with DoLP in the coarse-mode IRI is observed solely in Scenario III for D0P+L (22%).

The accuracy in AOD retrievals with DoLP is more pronounced in Scenario I (columns 9 and 10 in Table 7), for both aerosol modes (maximum reductions of 40% and 41% in the

fine and coarse modes of D1P+L, respectively), in which the aerosol loading is almost 0.40. This is in agreement with [28], who noticed that the effect of AOD errors decreases with higher AOD. In opposition to Scenario I, Scenario II (low aerosol loading) has the smallest RMSE reductions for its coarse mode (20% and 25% for D0P+L and D1P+L, respectively).

The SSA and LR (columns from 11 to 14 in Table 7) are the most positively affected properties when DoLP is considered in the GRASP retrievals because they are directly related to the scattered light and optical thickness. The higher RMSE reductions are in Scenario I (73% and 83% for the fine mode of D0P+L and D1P+L, respectively), followed by Scenario II (77% in the fine mode of D0P+L) and Scenario III with an error reduction of 20% in the fine mode of D0P+L.

The highest RMSE reductions of the LR are observed in Scenario I for the fine mode of D1P+L (40%), in the coarse mode of Scenario II with 54% and 76% (D0P+L and D1P+L, respectively), and in the coarse mode of Scenario III for the D1P+L combination (18%). The results of Scenarios I and III are in accordance with [15], who pointed out that the LR errors for the non-dominant mode showed reductions when the photometer and lidar are put together.

The SF retrievals (column 16 in Table 7) show significant reductions in the percentage of non-spherical particles of 71% (D0+L/D0P+L comparison) and 83% (D1+L/D1P+L comparison) for Scenario I, 21% (D0+L/D0P+L comparison) and 27% (D1+L/D1P+L comparison) for Scenario II, and 15% (D0+L/D0P+L comparison) and 14% (D1+L/D1P+L comparison) for Scenario III. In other words, the linear polarization of scattered light is less sensitive to spherical particles in all dominant modes. The small reduction in the averaged SF with DoLP can be related to the coarse particle contribution in Scenario III. The advantage of adding polarized information to retrieve SF is in agreement with the study of [7], who observed that the non-sphericity is reproduced more correctly by inversion with DoLP for dominant coarse mode.

The addition of more photometer channels with three-wavelength elastic lidar (D1+L) seems to supplement the information for retrieving some aerosol properties and reduce the RMSE for all aerosol scenarios. For instance, coarse-mode SSA (11%) and SF (43%) in Scenario I, coarse-mode RRI (7%), fine-mode IRI (29%), coarse-mode AOD (20%), fine-mode SSA (38%), both LR modes (5% and 12% in fine and coarse modes, respectively), and SF (25%) in Scenario II, coarse-mode SD (33%), coarse-mode RRI (67%), both IRI modes (35% and 11% in fine and coarse modes, respectively), both SSA modes (13% and 29% in fine and coarse modes, respectively), both LR modes (10% and 20% in fine and coarse modes, respectively), and SF (23%) in Scenario III. The RMSE results reinforce the improvements made with the use of DoLP in the GRASP retrievals.

## 4. Conclusions

The sensitivity of the GRASP code to the DoLP has been assessed. For the first time, this analysis explored the improvements in aerosol properties when using synergy between a polarized sun-sky-lunar photometer and a three-wavelength elastic lidar. The parameters have been inverted by means of synthetic data based on climatological values and realistic measurements performed by the Optical Remote Sensing group at the Universitat Politècnica de Catalunya in Barcelona, Spain. The inversions were carried out for three different aerosol scenarios (I—high AOD with dominant coarse mode, II—low AOD with dominant coarse mode, and III—high AOD with dominant fine mode), five synergy combinations (D0, D0+L, D0P+L, D1+L, and D1P+L), and two conditions to be compared with the reference (simulated data): (i) noise-free inversions and (ii) random noise inversions. Additionally, the fractional bias and correlation coefficient in the main layer have been calculated to better compare the VC profiles and the combinations with and without DoLP.

The GRASP algorithm fits the retrieved aerosol properties to the reference for the noise-free condition better than the retrievals without linear polarization. This is observed for all aerosol scenarios. Related to the comparisons between the random noise inversions (with and without DoLP) and the reference, some important conclusions were drawn and

organized in sequence: i and ii are about each aerosol mode; iii and iv are attributed to specific conclusions from Scenario II and III, respectively; v is with respect to SD and VC properties; vi is about the SF; and vii is related to the contribution of the D1+L combination for the inversions. They are listed below:

i. Even when the fine mode is the non-dominant one (Scenarios I and II), GRASP retrievals with DoLP contributions can significantly improve some of the microphysical and optical properties of that mode. For instance, maximum reductions are observed in the uncertainties of 29% (IRI from D0P+L in Scenario I), 8% (AOD from D0P+L in Scenario II), 56% (SSA from D1P+L in Scenario I), 36% (LR from D1P+L in Scenario I), 46% (RRI from D0P+L in Scenario II), and better adjustments of the averaged RN inversions to the reference. So, in coarse mode-dominated regimes, the inclusion in GRASP of the DoLP parameter helps to improve the retrieval of aerosol properties in the non-dominant, fine mode.

ii. The coarse mode of all scenarios presents gains with DoLP, mainly for D1P+L combination at longer wavelengths, which are expected because of the sensitivity of large particles to the polarized light when those particles are closer to their respective longer wavelengths (the opposite is also observed). The maximum reductions in the uncertainties are observed at 1020 and 1640 nm: 26% at 1020 nm and 15% at 1640 nm (RRI in Scenario I), 16% at 1640 nm (RRI in Scenario II), 28% at 1640 nm (LR in Scenario II), 36% at 1640 nm (RRI in Scenario III), and 16% at 1640 nm (LR in Scenario III). Thus, independently of the dominance or not of the coarse mode, the inclusion in GRASP of the DoLP parameter helps to retrieve the coarse mode of RRI and LR.

iii. The DoLP contributes to the accuracy of GRASP retrievals in Scenario II, dominant coarse mode with low aerosol loading ($AOD_{440}$ = 0.26), decreasing the overestimation and underestimation of the reference and reducing the uncertainties of the RN retrievals, with a maximum reduction of 46% (fine-mode RRI), 20% (coarse-mode RRI), 8% (fine-mode AOD), 7% (coarse-mode AOD), 2.5% (fine-mode SSA), 18% (coarse-mode SSA), 22% (fine-mode LR), and 28% (coarse-mode LR). This advantage can enhance the retrievals for low aerosol loading, in which better GRASP retrievals are expected for high aerosol loading [15,24,28,63].

iv. The use of DoLP in the retrieval of a dominant fine aerosol (Scenario III) displays an evident impact on the refractive indexes. The uncertainties reduce up to 30% and 23% for RRI (D0P+L and D1P+L, respectively) and 16% for IRI (D0P+L) at shorter wavelengths of the fine mode. Furthermore, the coarse-mode IRI has uncertainty reductions of 30% (D0P+L) and 10% (D1P+L), which is in opposition to the coarse-mode IRI in Scenarios I and II (no enhancements), confirming that the linear polarization is more sensitive to RRI than IRI for small particles [6]. In addition to the improvements in Scenario III, there are good fittings of the RN inversions to the reference for the properties in the coarse mode. Once again, DoLP contributes to improving the fine-mode estimations of the GRASP inversions.

v. For all scenarios, the addition of DoLP to the inversions has a minor contribution in the fine-mode SD and the coarse-mode VC.

vi. The SF accuracy is more notable for Scenarios I and II (dominant coarse aerosols), where the sensitivity of the non-spherical particles is more evident [7], decreasing the SF by 71% (D0P+L) and 83% (D1P+L) for Scenario I and 21% (D0P+L) and 27% (D1P+L) for Scenario II;

The addition of more photometer channels in the synergy with three-wavelength elastic lidar (D1+L) shows good estimations, reducing the RMSE mostly in the fine-mode RRI (29%) and coarse-mode SSA (11%) in Scenario I, the coarse-mode AOD (20%), fine-mode SSA (38%) and coarse-mode LR (12%) in Scenario II, the coarse-mode SD (33%) and coarse-mode RRI (67%), fine-mode IRI (35%), coarse-mode SSA (29%), and coarse-mode LR (20%) in Scenario III, and the SF in Scenarios I, II, and III (43%, 25%, 23%, correspondingly).

The RMSE confirms the enhancements with the D1+L combination on GRASP retrievals, mainly for Scenarios II and III.

Finally, the accuracy of the GRASP algorithm has also been investigated by the RMSE calculations, using the averaged RN inversions (with and without DoLP) and the reference for each aerosol property and their modes. The reductions are in line with the results found in the comparison between the aerosol properties inversions and the reference, mainly for the optical parameters (SSA and LR) when DoLP is considered in the inversions. This paper sheds light on the advantages that the combination of sun-sky photometers with polarization measurement capability and multi-wavelength lidars offers to improve the retrieval of aerosol microphysical and optical properties.

**Author Contributions:** Conceptualization, D.C.F.d.S.O., M.S., A.R.-G. and M.E.H.; Writing—Original Draft Preparation, D.C.F.d.S.O.; Writing—Review and Editing D.C.F.d.S.O., M.S., A.R.-G., C.M.-P., A.C., C.G.-D., S.L., O.D., A.L., M.E.H. and M.H.-G.; Software, O.D., A.L., M.E.H. and M.H.-G. All authors have read and agreed to the published version of the manuscript.

**Funding:** This research was funded by: the Spanish Ministry of Science and Innovation (grant no. PID2019-103886RB-I00); H2020 (grant nos. 654109, 778349, 871115 and 101008004); and Horizon Europe REALISTIC project (grant no. 101086690).

**Data Availability Statement:** Publicly available datasets were analyzed in this study. This data can be found here: https://aeronet.gsfc.nasa.gov (accessed on 18 September 2023) and https://scc.imaa.cnr.it/data_processing (accessed on 18 September 2023).

**Acknowledgments:** The authors are especially grateful to the Laboratoire d'Optique Atmosphérique at the Université de Lille and the GRASP-SAS company for supporting the secondments, and Ferrán Álvaro Saperas (IT support of the TSC) for installing the GRASP code in the UPC cloud.

**Conflicts of Interest:** The authors declare no conflict of interest. The sponsors had no role in the design, execution, interpretation, or writing of the study.

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
