# Peer review of "Evaluation of the Accuracy of the Aerosol Optical and Microphysical Retrievals by the GRASP Algorithm from Combined Measurements of a Polarized Sun-Sky-Lunar Photometer and a Three-Wavelength Elastic Lidar"

_remotesensing, doi:10.3390/rs15205010_

Round 1
Reviewer 1 Report
General comments:
The manuscript entitled ‘Evaluation of the accuracy of the aerosol optical and microphysical retrievals by GRASP algorithm from combined measurements of a polarized sun-sky-lunar photometer and a three-wavelength elastic lidar’ is well-written and scientifically sound. I recommend publication of the paper after addressing the following comments.
Specific comments:
The authors should mention about the configurations of particle shapes (more than just the sphere fraction) in both forward and inversion model in GRASP.
Minor comments / typos:
Line 203, ‘plan’ should be ‘plane’.
Line 205, a right bracket is missing.
Line 223-224, in lidar community, we usually use beta to denote backscatter, use alpha to denote extinction, and use delta to denote depolarization ratio.
Table 3, Telephoto lens for depolarization measurements, ‘523’ should be ‘532’.
Line 246, the equation is incorrect.
Table 4, first line, the variable should be specific for aerosol vertical profile, is it volume concentration or something else?
Table 4, last line, should be ‘Aerosol Absorption Profile’
Line 351, cannot find section 6.1.
Author Response
All issues answered in attached file

Reviewer 2 Report
The authors present an analysis of the sensitivity of the GRASP algorithm to the degree of linear polarization. Aerosol properties has been improved by exploiting the synthetic data between a polarized sun-sky-lunar photometer and a three-wavelength elastic lidar. It is a nice and well-organized paper with a clear focus. In my opinion, it can be published in this journal considering minor revisions.
Line 98-119: It is recommended that names be added when citing previous research. For example, the sentence in line 98 could be revised as: “Lopatin et al. (2013) explored the scope and the accuracy of the aerosol retrieval from”.
Line 337-340: How to define high or low aerosol loading? Why give these value in simulation test?
It will be better if the author give further analysis that the improvement of retrieval accuracy mainly comes from the contributions of short wave DoLP or longer wave DoLP.
Author Response
All issues answered in attached file

Reviewer 3 Report
Dear Authors,
I found the presented manuscript to be very interesting. I have found no issues with the applied methodology. The quality of the presentation requires some improvements, however, almost exclusively in the first two sections - please see the specific comments. I recommend the manuscript for publication after these minor issues are addressed.
Specific comments:
line 26: There is a problem with the structure of this sentence. What does "being possible" refer to? I would recommend splitting this sentence into two for better flow.
line 52: The interpunction is missing in this sentence. Please, replace "and also" with "but also". Also, consider starting a new sentence at "therefore".
line 69: what does "it" refer to?
line 83: "to" is missing before "the amount"; "... and to derive extended ..." - this is an improper structure of the sentence, please reformulate.
line 87: please change "over all" to "overall"
line 93: please change "frame work" to "framework"
line 97: "surface properties various observations" - something is missing here
line 102: "to conventional intensity only AERONET processing." - again, something is missing
line 105: please replace "when" with "where"
line 131: "at faster pace" - faster than what? Maybe use "fast"?
line 199: "level 1.5" - It was previously stated that level 2.0 AERONET data were used. Is it level 2.0 for older data and level 1.5 for data with polarization? Please clarify.
line 208: "The first UPC/EARLINET lidar system, operative since 1996, was performed by elastic channels at 532 nm and 1064 nm" - Lidar system was performed by channels? Please clarify.
line 225: "AERLINET" - Please change to EARLINET
line 252: "10-2 to 10-5" - Please add superscript
line 268: "According to [5,21,58], biomass fine aerosol particles polarize the scattered light, while the polarization by coarse non-spherical desert dust particles is weak." - And yet in fig 1 coarse fraction polarizes light stronger up to 120 deg. Please explain/elaborate.
line 283: "The Numerical Inversion module..." - the use cases alone do not explain the differences between single- and multi-pixel retrievals. Both approaches can be used for ground-based measurements, for instance (multi-pixel in the time domain). Please expand the explanation.
line 287: "carries out the simulations of the inverted remote sensing observations to simulate data" - What is simulated here, inversions or the measurements? The latter seems to be the case. Please confirm and refraze for clarity.
line 323: "eliminating the averages greater than 20 %" - Please clarify. Should it be "average differences" or similar?
line 365: "The covariance matrix." - Incomplete sentence
line 459: Please add "in" after "absent"
Figure 4: The layout suggests that all the quantities are unitless. LR is technically in steradians, though.
Figure 5: What is the reasoning behind using μm3/μm3 as a unit here (instead of just dimensionless or e.g. mm3/m3)? Is it aimed at better correspondence with the AERONET's μm3/μm2 unit used for the columnar volume size distribution?
line 613: "in which contributes overestimating the reference in this region" - Unclear, please refraze.
Sections 1 and 2: Interpunction is missing or incorrect in many sentences. Moreover, some sentences have unusual word ordering that makes them difficult to follow.
Author Response
All issues answered in attached file
